# MASK-GUIDED VIDEO GENERATION: ENHANCING MOTION CONTROL AND QUALITY WITH LIMITED DATA

## ABSTRACT

Recent advancements in diffusion models have brought new vitality into visual content creation. However, current text-to-video generation models still face challenges such as high training costs, substantial data requirements, and difficulties in maintaining consistency between given text and motion of the foreground object. To address these challenges, we propose mask-guided video generation, which requires only a small amount of data and is trained on a single GPU. Furthermore, to mitigate the impact of background interference on controllable text-to-video generation, we utilize mask sequences obtained through drawing or extraction, along with the first-frame content, to guide video generation. Specifically, our model introduces foreground masks into existing architectures to learn region-specific attention, precisely matching text features and the motion of the foreground object. Subsequently, video generation is guided by the mask sequences to prevent the sudden disappearance of foreground objects. Our model also incorporates a first-frame sharing strategy during inference, leading to better stability in the video generation. Additionally, our approach allows for incrementally generation of longer video sequences. By employing this method, our model achieves efficient resource utilization and ensures controllability and consistency in video generation using mask sequences. Extensive qualitative and quantitative experiments demonstrate that this approach excels in various video generation tasks, such as video editing and generating artistic videos, outperforming previous methods in terms of consistency and quality[1].

## 1 INTRODUCTION

In recent years, diffusion-based generative models (Ho et al., 2020; Song et al., 2020a;b) have made significant progress in text-to-image generation. Models such as DALLE2 (Ramesh et al., 2022), Stable Diffusion (Rombach et al., 2022), and Imagen (Saharia et al., 2022) have demonstrated the ability to generate diverse and high-quality images guided by text prompts. Given the success of text-to-image generation, some researchers have begun to explore applying these successful experiences to the field of text-to-video (T2V) generation. However, unlike image generation, video creation demands not only accurate alignment between text and individual frames, but also coherence between the text and the motion of the foreground object.

Some researchers have attempted to achieve text-to-video generation by training on a large amount of video data (Blattmann et al., 2023; Esser et al., 2023). While this method may make some progress, it requires a significant amount of computational resources and human effort for data and text annotation, making it impractical for real-world applications. To address this issue, Tune-A-Video (Wu et al., 2023a) introduced a new single-video fine-tuning setup for T2V generation. This approach significantly reduces the training workload, as it only requires the fine-tuning of a single video to complete the training on a consumer-grade GPU. However, Tune-A-Video (Wu et al., 2023a) tends to overfit the given video, making it highly susceptible to the background of the provided video.

---

[1]We will make our code publicly available once the paper is accepted.

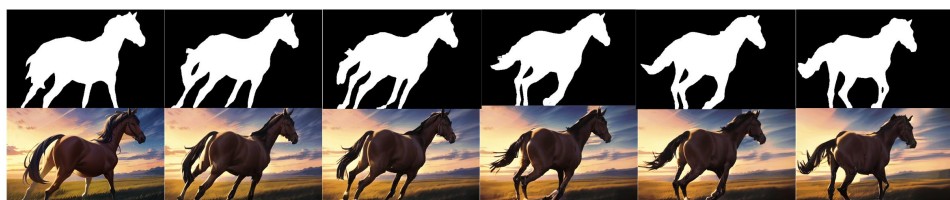

Horse run / A horse runs on the grass.

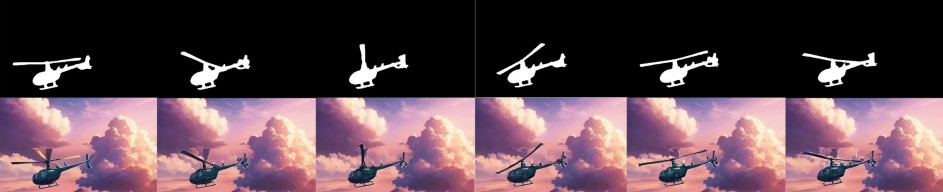

Helicopter / A helicopter flies in the pink sky.

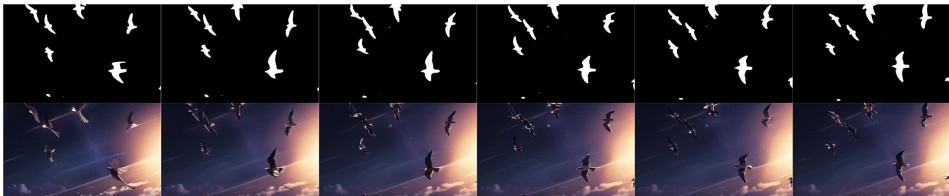

Birds fly / Birds fly in the dark sky.

Figure 1: Our model generates various videos consistent with the foreground mask and text prompts, delivering satisfactory results. The first frame is generated using the mask through ControlNet's Scribble model, capturing most of the content in the video frames. Subsequently, video generation is guided by the mask sequence, enabling effective control over text positions and motion regions while accurately distinguishing foreground and background, thereby preventing object disappearance.

To address above issues, LAMP (Wu et al., 2023b) is proposed and it keeps the first frame unchanged while learning subsequent frames, allowing the video diffusion model to focus mainly on motion learning. This approach effectively improves video quality and the degree of freedom in generation. Despite the progress made by LAMP (Wu et al., 2023b), there are still two challenges(see Fig. 2), which can be summarized as follows: (i) Imprecise Foreground Positioning and Motion Capture: The generative model encounters difficulties in accurately determining the foreground position specified by the text. This imprecision leads to subsequent errors in interpreting the movement of foreground objects, ultimately compromising the video's overall visual fidelity. (ii) Unnatural fusion between moving foreground and background: Even when the position and motion specified by the text are accurately captured, the fusion between the text-specified moving foreground and the background is still challenging. This unnatural fusion makes the model unable to distinguish between the text-described subject and the background, causing disappearance of foreground object.

To address these issues, introducing controllability to video generation is a potential solution. We propose a novel mask-guided video generation method for producing high-quality and controllable videos. Unlike previous works (Wu et al., 2023b;a), we introduce a foreground mask branch in the network to adjust attention for foreground positioning and motion capture. As shown in Fig. 2, our model effectively captures the foreground position and motion trajectory defined by the text. During inference, we can guide the motion generation according to the provided mask sequence, effectively distinguishing between the foreground and background and preventing the sudden disappearance of the foreground. This effectively controls the motion generation of the foreground extracted from the first frame. Following this approach, we can use a frame from the generated video as the first frame for the generation of subsequent frames, resulting in a longer video guided by the mask. Additionally, we use the latent features of the first frame as shared noise to improve the quality and stability of video generation. We conduct experiments on motion in various scenarios, and the

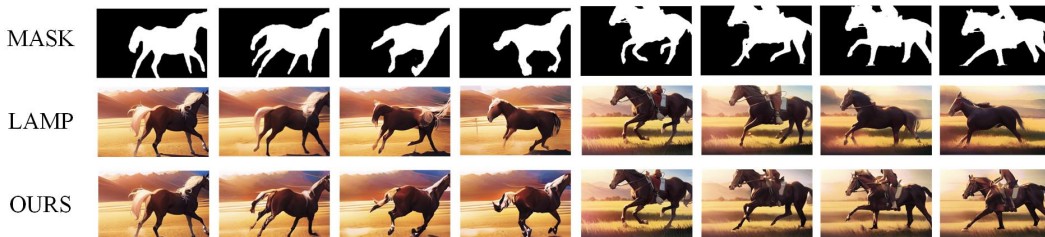

A horse is running on the grassland.    A person is riding a horse on the grassland.

Figure 2: From the comparison results, it is easy to see that in LAMP, the model does not accurately capture the direction of the horse's movement and failed to effectively distinguish the foreground from the background, resulting in the disappearance of the person. In contrast, our model has successfully addressed these issues, accurately capturing the position corresponding to the text and clearly distinguishing the foreground from the background.

results show that our method significantly improves the quality of generated videos. Our method can generate both short and long videos with training conducted in a single GPU, as shown in Fig 1.

In summary, our contributions are as follows: **(i)** This text-to-video method can be trained in a single GPU using only a small video dataset. **(ii)** Our model introduces an innovative mask-aware attention layer during training, which allows for more precise capture of the foreground region. During generation, by providing a mask sequence to control the video generation, it effectively prevents the blending of foreground and background, ensuring clear separation and stable presentation of the foreground in the video. **(iii)** We have empirically demonstrated that our model achieves excellent results in both consistency and quality. As a byproduct of using the first-frame conditional generation strategy, our method is capable of generating long videos while ensuring the continuity and consistency of the target object's motion.

## 2 RELATED WORKS

### 2.1 TEXT-TO-IMAGE GENERATION

In recent years, text-to-image generation technology has made significant progress in producing high-quality and semantically consistent images. Particularly, diffusion models (Ho et al., 2020; Song et al., 2020a;b) have been widely favored over GANs (Goodfellow et al., 2020; Zhang et al., 2017; Xu et al., 2018) and VAEs (Kingma & Welling, 2013; Sohn et al., 2015; Van Den Oord et al., 2017) for their excellent text consistency, high-quality generated images. Models such as GLIDE (Nichol et al., 2021), Imagen (Saharia et al., 2022), DALL·E2 (Ramesh et al., 2022), and LDM (Rombach et al., 2022) enhance the association between text and images using the CLIP (Radford et al., 2021) model and generate images with fine textures and accurate semantics through powerful image generation architectures like Transformers (Vaswani et al., 2017). Additionally, to achieve a more personalized generation, researchers have developed methods like DreamBooth (Ruiz et al., 2023) and textual inversion (TI) (Gal et al., 2022), which expand the tokenizer's vocabulary by embedding user-provided concepts in the text embedding space, and improve image generation through denoising processes. Furthermore, the ControlNet (Zhang et al., 2023) has brought new vitality to the research on text-to-image generation by introducing more controllable input conditions to Stable Diffusion, such as depth maps, poses, and so on. In our method, we use the first frame mask to create the first frame of the video by ControlNet (Zhang et al., 2023), which is then used to generate the subsequent frames.

### 2.2 TEXT-TO-VIDEO GENERATION

The success of text-to-image generation has inspired researchers to explore the field of text-to-video generation, leading to some notable achievements, such as ImagenVideo (Ho et al., 2022) developed by Google Research, Make-A-Video (Singer et al., 2022) by Meta AI, MagicVideo (Zhou et al.,

2022), VideoComposer (Wang et al., 2024)], CogVideo (Hong et al., 2022), and AnimateDiff (Guo et al., 2023). These models have achieved a transition from generating static images to dynamic videos through multimodal learning, diffusion models, and keyframe generation, as well as cognition and understanding. However, these models currently focus primarily on generating short videos. To address the challenge of generating long videos, existing methods typically rely on autoregressive models and diffusion models. Autoregressive models like NUWA-Infinity (Wu et al., 2022), Phenaki (Villegas et al., 2022), and TATS (Ge et al., 2022) generate long video content by using generated frames as conditions for subsequent frames. On the other hand, diffusion models like MCVD (Voleti et al., 2022), FDM (Harvey et al., 2022), PVDM (Yu et al., 2023), and LVDM (He et al., 2022) also adopt a similar autoregressive mechanism, generating coherent video sequences by creating high-quality intermediate frames and gradually interpolating them. However, these models undeniably require substantial training resources, while methods like Text2Video-Zero (Khachatryan et al., 2023) and ControlVideo (Zhang et al., 2024b) employ zero-shot techniques, which do not require fine-tuning but depend on pre-trained models and large-scale pre-training data, demanding high data quality and diversity. In specific domains or scenarios, these methods may fail to generate sufficiently detailed and accurate content due to the lack of specialized training on domain-specific data. Additionally, methods like Control-A-Video (Chen et al., 2023) and MoonShot (Zhang et al., 2024a) draw on the ideas of ControlNet (Zhang et al., 2023), using depth maps, edges, and other motion sequences to generate videos. While these methods can improve generation quality, they also require considerable training resources. To address this issue, we follow the approach of the LAMP (Wu et al., 2023b) model, which can be trained with only a small number of samples and a single GPU. Different from their approach, we introduce a method that incorporates foreground masks during training, allowing for more precise capture of the foreground region. This effectively prevents the blending of the foreground and background, ensuring clear separation and stable presentation of the foreground in the video.

## 3 METHOD

In this section, we will first briefly introduce our training pipeline in Sections 3.1 . Then, in Section 3.2, we will detail the proposed mask-guided video generation method. Our method uses the first frame as a condition, effectively decoupling mask-guided frame generation and motion generation, thereby reducing training costs. We also introduce a foreground mask branch, incorporating the foreground mask into the network to adjust attention for foreground positioning and motion capture, making the model focus more on the text-specific foreground region, making the model focus more on the foreground positions matching the text. During the inference phase, the mask sequence enables the model to effectively distinguish between the foreground and background, preventing the disappearance of foreground objects. In Section 3.3, we will provide a detailed explanation of the proposed mask-aware attention layer. Additionally, in Section 3.4, we will briefly introduce our first-frame shared sampling strategy, which improves the quality and stability of video generation.Details about the theoretical background of Latent Diffusion Models (LDMs) (Rombach et al., 2022) and their relevant applications can be found in Appendix A.

### 3.1 TRAINING PIPELINE

Current text-to-video methods are commonly trained on large-scale datasets. Moreover, they are not only computational expensive. However, this approach is not only computationally expensive but also faces significant challenges in capturing the motion patterns of foreground objects, especially in achieving precise control over motion trajectories. To address these issues and enhance the controllability of foreground motion, we optimize the LAMP (Wu et al., 2023b) model and propose a new video generation pipeline.

Our model first processes a video set $V = \{V_i | i \in [1, n]\}$ and uses the Segment Anything model (Kirillov et al., 2023) to extract foreground masks from the videos, generating a corresponding set of foreground mask videos. The training data includes $n$ videos and their foreground mask video sets, along with a text $P_m$ describing a shared motion pattern. By fine-tuning a pre-trained T2I model based on the given video set and motion prompt, the model can generate new videos $V'$ with motion patterns specified by prompt $P_m$.

In our pipline, the model focuses on learning the common motion patterns shared across a small set of videos while ignoring unimportant details. Hence, our model can be trained on a small number of videos. Additionally, our model can more accurately capture foreground motion, achieving fine-grained control over motion trajectories.

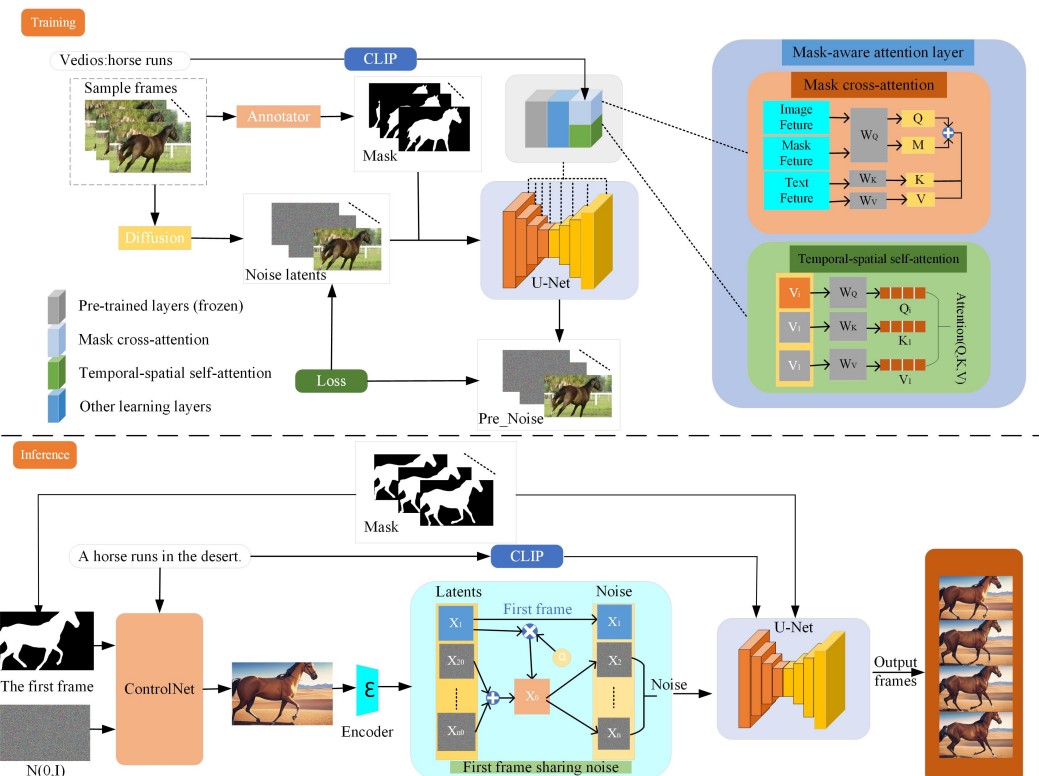

Figure 3: Overall framework of our mask-guided video generation method. We apply trainable temporal-spatial self-attention and mask cross-attention within the U-Net, enabling the model to focus more on the foreground. **Training:** The input consists of randomly sampled video frames, text prompts describing their motion pattern, and mask sequence maps extracted from an annotator. We add motion-prior noise to every latent signal except for the first frame, and the model is trained to predict the subsequent noise conditioned on the first frame. **Inference:** The first frame of the motion sequence mask is generated using ControlNet and the text prompt. After training, the generated first frame can be used to generate subsequent frames with content priors.

## 3.2 MASK-GUIDED TRAINING

Based on our observations, the first frame of a video often contains key information of the entire video. Therefore, we use the content of the first frame as a condition to generate the subsequent frames. This approach not only reduces the amount of training content but also allows the model to focus more on learning the motion in the video. During training, we keep the first frame unchanged and continuously add noise to the other frames to capture the motion in the subsequent frames based on the first frame. As shown in Fig 3, we represent the sampled $n$ video frames as $V = \{f_i | i = 1, ..., n\}$. These frames are embedded into latent space using an encoder and we obtain $X = \{x_i | i = 1, ..., n\}$. We then keep the first frame $x_1$ unchanged and apply a forward diffusion process to the subsequent frames $x_2, ..., x_n$ to obtain the noisy video frame sequence $\varepsilon_2, ..., \varepsilon_n$. The loss function can be expressed as:

$$L = E_{x,\varepsilon \sim N(0,I),t,c_p} \left[ \left\| \varepsilon_{2:n} - \varepsilon_{2:n}^{\theta}(x_t, t, c_p) \right\|_2^2 \right],\tag{1}$$

where $\varepsilon_{2:n}$ represents the noisy video frame sequence from the 2nd to the $n$-th frame and $\varepsilon_{2:n}^{\theta}$ and is the noise predicted by the neural network at time step $t$ conditioned on the input $c_p$ (e.g., text

prompts). This way, we retain the original signal of the first frame, and the loss function is only applied to the frames from the 2nd to the $n$-th frame. Using this method, the model gains the capability to generate a video with the motion pattern of the video set according to the first frame.

During inference, we input the first frame $m_1$ of the mask motion sequence, the text prompt $c_p$, and random Gaussian noise $x_1$ into the ControlNet (Zhang et al., 2023) model to obtain the initial frame $v_1$:

$$v_1 = \text{ControlNet}(x_1, c_p, m_1), \tag{2}$$

After obtaining the desired first frame, we encode the first frame and obtain $E(v_1)$. The subsequent motion sequence frames are generated using:

$$v = \text{MaskVideo}\left(x, c_p, m, E(v_1)\right), \tag{3}$$

The video generation method we developed, based on the first-frame condition, offers the following advantages: First, it allows precise control over the content of dynamic videos, enabling content diversification by adjusting the first frame. Second, this method excels in generating long videos, as we can use any frame from the generated video as a new starting point, enabling iterative extension of the video, thus overcoming the limitation of video length. This distinguishes our approach from other techniques that can only generate short videos in a single step. Specifically, the proposed algorithm is outlined in Alg 1.

---

**Algorithm 1** Mask-Guided Video Generation

**Input**: Mask $M = \{m_1, m_2, ..., m_n\}$ ,Text prompt $c_p$
**Parameter**: $T$
**Output**: $V$:generated video
 1: $V_1 = ControlNet(\varepsilon_1, c_p, m_1)$
 2: $\varepsilon_s = E(V_1)$
 3: **for** $i = 2$ to $T$ **do**
 4:     $\epsilon_i = \alpha\epsilon_s + (1-\alpha)\epsilon_i$
 5: **end for**
 6: **for** $t = 2$ to $T$ **do**
 7:     $v_t = \text{MaskVideo}(c_p, M, \epsilon_t, \epsilon_s)$
 8:     $v.\text{append}(v_t)$
 9: **end for**
 10: $V = D(v)$
 11: **return** $V$

---

### 3.3 MASK-AWARE ATTENTION LAYER

To integrate with the proposed pipeline and facilitate subsequent frames referencing the conditions established by the first frame, we introduce a new temporal-spatial self-attention. To ensure consistency, all key and value features are derived from the first frame. More specifically, each self-attention layer takes as input a feature map $v_i$ and projects it linearly into query, key, and value features as follows:

$$Q = w^Q \cdot v_i, \quad K = w^K \cdot v_1, \quad V = w^V \cdot v_1, \tag{4}$$

where $Q, K, V \in R^{BNS \times HW \times \frac{C}{S}}$. Here, $B$ is the batch size, $N$ is the number of frames, $H$ is the height, $W$ is the width, $C$ is the number of channels, and $S$ is the number of attention heads. The attention scores of the self-attention layer are then obtained as follows:

$$\text{SelfAttn}(Q^i, K^1, V^1) = \text{Softmax}\left(\frac{Q^i(K^1)^T}{\sqrt{d}}\right)V^1, \tag{5}$$

where $i =\in 1, \ldots, n$ indicates that the extracted feature map comes from the $i - th$ frame, and $d$ is the hidden layer size. The attention scores establish the connection between the first frame and subsequent frames.

Additionally, to better match the foreground with the text, we propose mask cross-attention, which incorporates the mask into the calculation of the attention scores in cross-attention. Specifically, we

first project the the original feature map $v$ from the U-Net, the foreground mask $m$, and the text embedding vector $c$, using the following equations:

$$Q = w^Q \cdot v, \quad K = w^K \cdot c, \quad V = w^V \cdot c, \quad M = w^Q \cdot m, \tag{6}$$

where $Q, M \in R^{BNS \times HW \times \frac{C}{S}}, \quad K, V \in R^{BNS \times L \times \frac{C}{S}}$. The attention scores from the mask are then added to the original scores:

$$\text{CrossAttn}(Q, K, V, M) = \text{Softmax}\left(\frac{QK^T + MK^T}{\sqrt{d}}\right) V, \tag{7}$$

With this method, the degree of matching between text features and the motion of the foreground object is significantly improved, allowing the model to more accurately identify and capture the motion of the foreground object, achieving high consistency between text features and the motion of the foreground object. This not only enhances the quality of video generation but also effectively avoids difficulties in maintaining consistency between given text and its motion, making the generated results more natural and coherent in terms of visual effects.

### 3.4 FIRST-FRAME SHARED SAMPLING STRATEGY

During inference, most existing methods introduce a random shared noise that is fed into the model throughout the video generation process. This is because reducing the noise variance can narrow the dynamic range of the latent space, which contributes to a more stable generation process. However, we found that if the shared noise is a randomly sampled $\varepsilon_s \sim N(0, I)$, the generated results sometimes show a significant color loss in the foreground object and instability in the background ,as shown in Fig 4(a). To address this issue, we convert the first frame image into a latent embedding $\varepsilon_s$ and regard it as shared noise. Then, we sample a noise sequence $[\varepsilon_2, \cdots, \varepsilon_n]$ from the same distribution as the base noise of samples. The shared noise is then added to the subsequent frames at a certain ratio, specifically: $\varepsilon_i = \alpha\varepsilon_s + (1 - \alpha)\varepsilon_i$, where $\alpha$ is a balance parameter. According to our experiments, setting $\alpha$ to 0.2 yields the best results. Adding the first frame's features to the subsequent frames effectively prevents the loss of certain features during video generation and ensures continuity between frames. The result is shown in Fig 4(b).

## 4 EXPERIMENT

### 4.1 IMPLEMENTATION DETAILS

In our experiments, we train our model on only 5 to 8 videos with the same motion pattern, extracting the foreground masks using the "segment anything" method (Kirillov et al., 2023). In each iteration, we randomly sample a 16-frame clip from the original video and the mask video, with all frames resized to a resolution of $320 \times 512$ before being input into the U-Net network.We use the relatively lightweight SD-v1.4 (Rombach et al., 2022) for computationally intensive subsequent frame prediction, thereby reducing inference costs, and we train the model for 15,000 epochs. We only update

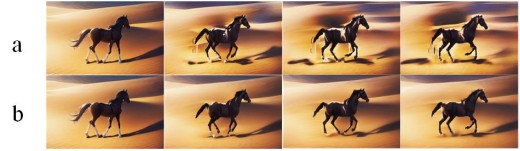

Figure 4: Given the text prompt "a horse runs in the desert," we conduct a comparison between using and not using the first-frame sharing strategy and the random noise sharing strategy.

the parameters of the newly added layers, as well as the parameters in the self-attention and cross-attention layers. The learning rate was set to $3.0 \times 10^{-5}$. During the inference phase, we use the ControlNet (Zhang et al., 2023) model to generate the first frame. Subsequently, by inputting the mask's motion sequence, the corresponding motion video can be generated. All experiments are conducted on a single NVIDIA RTX 4060Ti, requiring approximately 15GB of memory for training and around 12GB of memory for inference.

## 4.2 COMPARISON

Our method and baseline methods are trained on three types of data: animal movement, multi-object movement, and rigid body movement, as shown in Fig 1. The following methods are selected as our baselines: (i) Tune-A-Video (Wu et al., 2023a), a video editing model that enables high-quality, personalize video editing and special effects by adjusting and fine-tuning existing video content. (ii) Text2Video-Zero (Khachatryan et al., 2023), which utilizes the ControlNet (Zhang et al., 2023) model for video generation. It is based on ControlNet, and employs the first-only cross-frame attention on Stable Diffusion without finetuning. (iii) LAMP (Wu et al., 2023b), a framework for video generation base on a small amount of video data. It focuses on learning motion patterns by separating content and motion patterns, generating videos with higher degrees of freedom.

We present a comparison between our method and several baseline networks. The results are shown in Fig 5. The results indicate that Tune-A-Video (Wu et al., 2023a) still has room for improvement in terms of video quality and occasionally exhibits excessive adherence to the given video content. Text2Video-Zero (Khachatryan et al., 2023) demonstrates high sensitivity to the input video, with its generated output frequently being influenced by the background of the provided video. For example, the appearance of a fence in the background in Fig 5(c) is clearly unreasonable. Additionally, Text2Video-Zero heavily relies on pre-trained models, and when there is no fine-tuning with specific domain data, it tends to generate results with artifacts, failing to meet the needs for fine-grained video generation in specific domains. LAMP (Wu et al., 2023b) also generates videos based on the first frame, but it does not capture foreground objects well, leading to undesirable effects like the sudden disappearance of foreground objects. In contrast, as shown in Fig 5(e), our model effectively captures the text-specified motion trajectory and clearly distinguishes foreground from the background, avoiding the issue of the foreground suddenly disappearing.

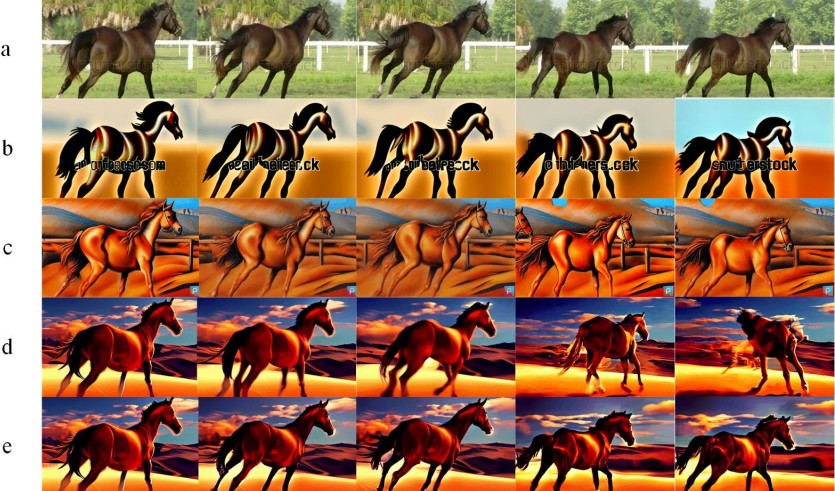

Figure 5: Comparison between our method and baselines when the text prompt is "A red horse runs in the desert". (a) Input video. (b) Tune-A-Video. (c) Text2Video-Zero. (d) LAMP. (e) Ours.

## 4.3 QUALITATIVE RESULTS

In this section, we evaluated our model and the baselines in terms of text-to-video alignment and inter-frame consistency.

**Objective metrics.** Text alignment assesses whether the generated video accurately reflects the description in the input text, ensuring a high match between the content and the text prompt. To achieve this, we employ the CLIP model to quantify text alignment by comparing the embedding similarity between the generated video frames and their corresponding text descriptions. The CLIP model maps both text and images into the same embedding space, allowing us to compute similarity scores between generated frames and text prompts, ultimately obtaining the average CLIP score (Radford et al., 2021) for each frame. Frame consistency measures the smooth transitions and co-

Table 1: Quantitative comparisons of text-to-video methods in frame consistency and textual faithfulness.

| Method | Frame Consistency | | Textual Faithfulness | |
|---|---|---|---|---|
| | CLIP Score↑ | User Preference↑ | CLIP Score↑ | User Preference↑ |
| Tune-A-Video | 88.9 | 11.0 | 29.4 | 12.4 |
| Text2Video-Zero | 96.1 | 9.4 | 30.2 | 12.4 |
| LAMP | 94.3 | 24.5 | 30.3 | 22.2 |
| OURS | **96.7** | **55.0** | **31.0** | **53.0** |

herence between video frames, ensuring that the generated video appears visually continuous and natural. High frame consistency indicates that movements and scene changes in the video are fluid, without abrupt transitions. We can use the CLIP model to evaluate consistency by comparing the similarity between frames.

In comparative experiments with baselines, we use the same motion sequence and provide five different scene prompts: "A red horse runs in the desert." "A white horse runs on the grassland." "A black horse runs on a flat snowy plain." "A grey horse runs in the pink sky." and "A brown horse runs on the road." We obtain five sets of results and average the scores for each. According to the experimental results in Table 1, our results outperform the other models in both text alignment and frequency consistency.

**User study.** We conduct a user evaluation survey to compare the performance of our method with other publicly available generation methods. Specifically, we create a questionnaire using 22 video samples, providing each evaluator with a set of text prompts and the corresponding generated results. We ask them to select the better generated videos based on two criteria: video quality and the alignment between the prompts and the generated videos. Ultimately, we received 33 completed questionnaires. As shown in Table 1, evaluators prefer our generated videos in both aspects. In contrast, Tune-A-Video, which only uses DDIM inversion for structural guidance, fails to produce consistent and high-quality videos, while the videos generated by Text2Video-Zero also exhibit lower quality.

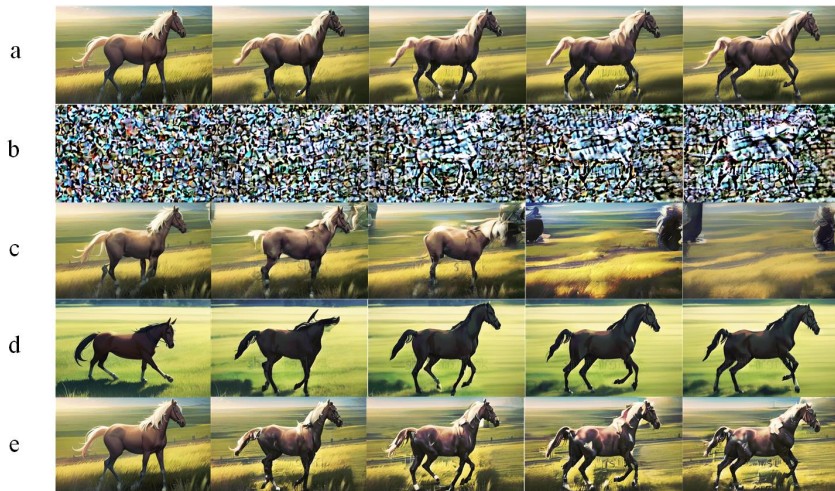

Figure 6: **Ablation results.** Given the text prompt "A horse runs on the grass," we conduct ablation experiments with the following variations: (a) our model; (b) without using the first frame for generation; (c) without providing the motion sequence mask; (d) without using ControlNet to generate the first frame; (e) without using the latent representation of the first frame as shared noise.

## 4.4 ABLATION STUDY

To validate the effectiveness of our method, we conduct ablation experiments by removing key components one by one to observe the impact on the quality of the generated videos. First, we completely remove the conditioning on the first frame and directly generated video frames from random noise. The resulting video, as shown in Fig 6(b), lost the content described by the text, demonstrating the foundational role of the first frame in generating the video. Secondly, we remove the motion sequence mask during inference. The resulting video, as shown in Fig 6(c), exhibited unclear motion patterns and poor frame-to-frame coherence, highlighting the importance of the mask sequence in guiding and capturing motion information. Thirdly, we did not use ControlNet (Zhang et al., 2023) to generate the first frame, instead generating a random initial frame. As shown in Fig 6(d), the generated first frame did not align with the first frame mask, leading to poor overall video generation quality. This further validates the advantage of using ControlNet (Zhang et al., 2023) for generating the first frame based on the mask. Finally, we replace the shared noise with random noise instead of using the latent representation of the first frame. The results, as shown in Fig 6(e), indicated a significant decline in video quality and consistency, with the foreground objects' colors becoming too bright and losing their original hues. This demonstrates the importance of shared noise. Through these ablation experiments, we demonstrate the crucial role of the motion sequence mask, shared noise, first frame results, and ControlNet (Zhang et al., 2023) in video generation. These components enable the model to effectively control the motion trajectory of the foreground text, prevent the issue of the foreground suddenly disappearing, and improve the quality of the generated video.

## 5 CONCLUSION

In this paper, we propose an innovative mask-guided video generation method that effectively addresses significant challenges in existing technologies, such as difficulties in maintaining consistency between foreground text and its motion. By introducing foreground masks during training, this method significantly improves the model's ability to learn foreground motion, thereby greatly enhancing the quality and controllability of the generated videos. Our approach can be trained on a small number of videos with a single GPU, utilizing ControlNet to generate the content of the first frame and guiding the generation of subsequent video frames through masks, achieving diversity and controllability in the motion of the foreground object. The experimental results show that this method not only improves the quality and consistency of the generated videos but also efficiently reduces training resource consumption.

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

## A  PRELIMINARY

In the preliminary knowledge section of this study, we will discuss the related concepts and applications of Latent Diffusion Models (LDMs). The main feature of LDMs Rombach et al. (2022) is operating in the latent space of an autoencoder, which provides significant computational advantages when generating high-quality images.

Given an input image $I$, its latent feature representation $x_0 = E(I)$ is first extracted through an encoder $E$. In the forward diffusion process, noise is gradually added to these latent features, with each noisy image at step $t$ represented by the following conditional probability distribution:

$$q(x_t|x_{t-1}) = N\big(x_t; \sqrt{\alpha_t}x_{t-1}, (1 - \alpha_t)I\big), \tag{8}$$

where $\alpha_t$ is a hyperparameter controlling the noise intensity at step $t$, and $q(x_t|x_{t-1})$ is the conditional probability distribution for generating $x_t$ given $x_{t-1}$. Additionally, we can directly sample $x_t$ at any time step t from $x_0$ as:

$$q(x_t|x_0) = N\big(x_t; \sqrt{\bar{\alpha}_t}x_0, (1 - \bar{\alpha}_t)I\big), \tag{9}$$

where $\bar{\alpha}_t = \prod_{i=1}^{t} \alpha_i$ , representing the cumulative diffusion coefficient.

As noise is added, the information of the initial features $x_0$ gradually deteriorates with increasing time steps t, and eventually, $x_t$ approaches a standard Gaussian distribution. To reconstruct $x_0$ from random noise, we use a U-Net structured neural network $\varepsilon_\theta$ trained to estimate the noise added in the forward process. Specifically, during reverse inference, the deterministic DDIM (Denoising Diffusion Implicit Models) sampling method is used to progressively generate $x_{t-1}$ from $x_t$. The formula is as follows:

$$x_{t-1} = \sqrt{\bar{\alpha}_{t-1}}\hat{x}_0 + \sqrt{1 - \bar{\alpha}_{t-1}}\varepsilon_\theta(x_t, t, c_p), \tag{10}$$

where $\hat{x}_0$ is the predicted value of $x_0$ corresponding to time step $t$, obtained $\hat{x}_0 = \frac{x_t - \sqrt{1-\bar{\alpha}_t}\varepsilon_\theta(x_t,t,c_p)}{\sqrt{\bar{\alpha}_t}}$ and $\varepsilon_\theta(x_t, t, c_p)$ is the noise predicted by the neural network at time step $t$ conditioned on the input $c_p$ (e.g., text prompts).

The training objective is to minimize the difference between the noise predicted by the model and the actual noise. The loss function can be defined as:

$$L_{simple} = E_{x_0,\varepsilon,t}\left[\|\varepsilon - \varepsilon_\theta(x_t, t, c_p)\|^2\right], \tag{11}$$

where $\varepsilon$ is the noise from a standard normal distribution, and $\varepsilon_\theta$ is the noise predicted by the neural network.

In the final inference stage, we sample noise $x_t$ from a standard Gaussian distribution $N(0, I)$ and progressively derive $x_0$ under the guidance of text prompts $c_p$ using the DDIM Song et al. (2020a) sampling method. Finally, we use a decoder $D$ to convert the obtained latent features $x_0$ into the final generated image $I' = D(x_0)$.

## B  MORE VISUALIZATIONS

We focused on three types of image sequences: animal motion, multi-object motion, and rigid body motion, as shown in Fig 1. The corresponding text prompts were "A horse runs on the grass," "A helicopter flies in the pink sky," and "Birds fly in the dark sky." We first generated the initial frame using ControlNet's Scribble model to ensure the accuracy of the motion content. Then, the video content was generated by guiding it with hand-drawn or extracted mask sequences. Our model achieved excellent results in terms of content consistency.

## C  LONGER VIDEO GENERATION

We employ an autoregressive generation approach to achieve long video generation using only a small amount of training videos on a single GPU. In Fig 7, we showcase the results of long video generation. Given a motion sequence of 24 frames, we need to generate a video based on the prompt "A horse runs across a flat desert plain under a midday sun in a pop art painting style.". We use a

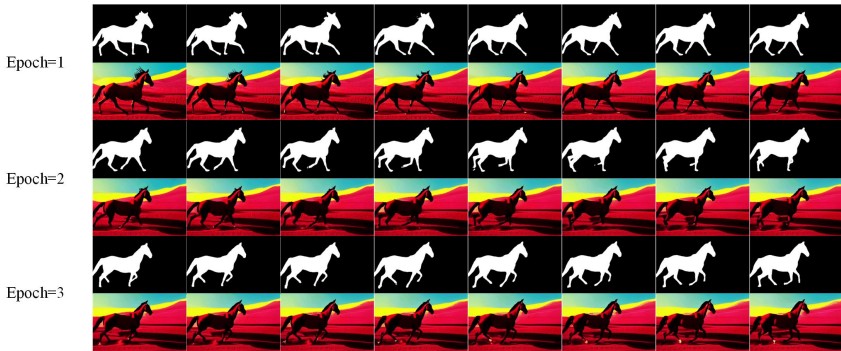

Figure 7: **Auto-Regressive Generation.** Our model is capable of generating long videos. Given the text prompt "A horse runs across a flat desert plain under a midday sun in a pop art painting style," the video frames were generated using a first-frame-based method, producing a 24-frame video after three epochs.

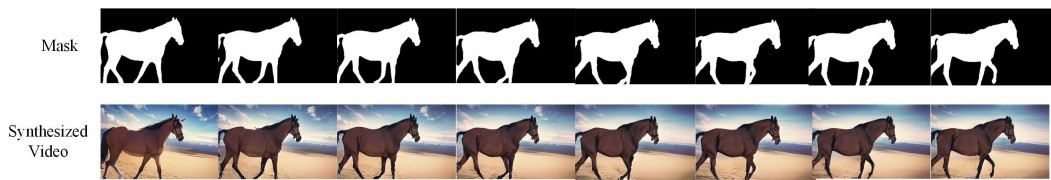

Figure 8: **Limitation of visualization.** Mask-guided video generation struggles to produce videos beyond the input motion sequence. The text prompt does not match the given mask motion sequence, and the resulting video is less than satisfactory, which reduces the overall quality and consistency of the video.

three-step generation process, where in each step, we generate 16 frames. Then, we select the ninth frame from the generated results as the new first frame for the next generation process. Finally, the generated frames are assembled according to their respective sequence positions. The results demonstrate excellent foreground consistency, indicating that our method is effective for generating longer videos.

## D  LIMITATION

Although our mask-guided video generation method achieves a high degree of consistency between the motion trajectory of the foreground object and the text description, the method still has certain limitations when generating videos beyond the scope of the input mask motion sequence. For example, when the input motion mask shows a horse walking, even if the text prompt is "a horse running in the desert," the generated video still strictly follows the given mask motion sequence, as shown in Fig 8. While the background may display a motion trend, the foreground horse still maintains a walking posture. Therefore, when there is a clear conflict between the text prompt and the motion sequence, the generated video tends to prioritize the input motion sequence, ignoring the dynamic information implied in the text prompt. To address this issue, future research will focus on how to adaptively adjust the motion sequence based on the text prompt, allowing the generated video to not only better align with user expectations but also exhibit greater dynamic expressiveness.

