# OpenReview forum: "Mask-Guided Video Generation: Enhancing Motion Control and Quality with Limited Data"
_ICLR.cc/2025/Conference — ICLR 2025 Conference Withdrawn Submission_

### Official Review · Reviewer_nq54 · 2024-10-24

**Soundness:** 2
**Presentation:** 2
**Contribution:** 2
**Rating:** 3
**Confidence:** 5

**Summary:**

This paper presents an image-to-video generation method with mask guidance. Thanks to this design, the method can generate video with very limited data. This method needs frame-specific masks for training and testing. The method also has a first frame-sharing noise method to enable better temporal consistency and a mask-aware attention model. This method compares several zero-shot/one-shot methods.

**Strengths:**

- Mask-guided is novel for me as a video generation condition.
- This method requires only a small dataset for training.

**Weaknesses:**

- I am curious about the motivation behind this paper. The mask is hard to generate when inference, which makes this method impractical.
- There are no video results for this paper.
- The comparison methods are too old to evaluate the performance of this method.
- Generating videos using small datasets from stable diffusion (text-to-image model) is out of fashion. Current state-of-the-art methods directly generate videos from large-scale training.

**Questions:**

1. Why do we need mask-aware video generation? How to get diverse results when the mask is provided.
2. Where is the video demo?
3. There is only a visual ablation of this method in the paper, what about the numerical results in the larger scale?

---

> ### Author Response · Authors · 2024-11-18
>
> 1. The motivation behind mask-aware video generation lies in addressing the accuracy and consistency of foreground object motion, especially in scenarios where text-to-video control demands are high, such as video editing and virtual character control. By utilizing mask-aware generation, this study enables precise control over the motion trajectories of foreground objects during video generation, ensuring that they neither disappear unexpectedly nor become confused with the background. Through the manipulation of different mask sequences, users can generate videos with diverse motion patterns.
> 2. A portion of the generated results will be uploaded as supplementary materials.
> 3. This study offers a lightweight solution for scenarios with small datasets, making it suitable for applications under resource-constrained or scenario-specific conditions. While mainstream methods typically rely on models trained on large-scale datasets, small dataset generation approaches remain valuable for customized tasks requiring specific foreground control in video generation.

---

### Official Review · Reviewer_iFVu · 2024-11-02

**Soundness:** 2
**Presentation:** 2
**Contribution:** 1
**Rating:** 3
**Confidence:** 5

**Summary:**

This paper introduces a mask-guided video generation approach. By integrating mask sequences that isolate the foreground object and employing a first-frame sharing strategy, the model can control object motion and ensure stability across frames. The method requires retraining a new model on each set of videos of new motion concepts.

**Strengths:**

1. The showcased videos have high quality, benefiting from the pre-trained text-to-image generation model.
2. This paper is technically clear. It provides detailed descriptions of the proposed method and implementation details.

**Weaknesses:**

1. The task setting in this paper is very weird. Training a video generation model on a small set of videos with a shared motion concept has already been addressed in prior works like Tune-A-Video and LAMP. This paper’s approach differs by introducing masks as a strong constraint to guide video generation, which significantly limits the diversity of generated videos. Consequently, users must not only train a new model for each video set but also should supply a mask sequence, combining the drawbacks of few-shot training and controllable video generation. This approach requires computing resources to retrain new models, with the generated video diversity strictly constrained by the provided mask sequence. It is suggested to provide a more detailed comparison of computational requirements and generated diversity between the proposed method and prior approaches, such as Tune-A-Video and LAMP.

2. It is difficult to attribute the final quality of motions in the generated videos to either the model's learned motion concept or the reduced search space resulting from the mask sequence, which could be driving motion stability. The lack of an ablation study on this point leaves the source of these improvements unclear. It is suggested to add specific ablation experiments that would help isolate the contributions of the learned motion concept versus the mask sequence constraints.  Adding specific ablation experiments to isolate the contributions of the learned motion concept from the constraints imposed by the mask sequence is recommended.

3. The technical contributions of this paper are limited, as it merely combines LAMP with mask-guided generation in a straightforward manner. Few-shot motion learning and mask-guided controllable generation have already been extensively explored in prior works.

4. The experimental results and analysis are quite limited. The quality of motion generation is a crucial aspect, yet there is a limited analysis of this in the experiments, and quantitative results, including those in Table 1, lack any evaluation specific to motion. Furthermore, mask guidance is a core component of the proposed method that significantly influences the generated outcomes, but the proposed method is not compared against other mask-guided approaches. It is suggested to add a metric that focuses on the motion quality. The proposed method is suggested to be compared with some mask-guided methods, like FateZero.


[1] Qi, Chenyang, et al. "Fatezero: Fusing attentions for zero-shot text-based video editing." Proceedings of the IEEE/CVF International Conference on Computer Vision. 2023.

**Questions:**

1. The authors are suggested to cover a more comprehensive discussion of existing methods and reorganize the related works section on text-to-video generation by splitting it into two parts: few-shot/zero-shot motion learning and mask-guided generation. This structure would provide an important context for understanding the novelty and positioning of this paper.

---

> ### Author Response · Authors · 2024-11-18
>
> 1. This study primarily focuses on video generation scenarios with limited samples that require strong foreground control. The goal is to achieve controllable generation under constrained resources, with a priority on ensuring video consistency and foreground controllability.
> 2. Unlike Tune-A-Video and LAMP, our mask-guided approach makes trade-offs in generation diversity but provides greater robustness in terms of consistency and foreground control. We observed that while LAMP offers higher generation freedom, it often results in issues such as missing foregrounds and uncontrolled motion directions. To address this, we introduced a control signal (mask), sacrificing some diversity to achieve higher consistency.
> 3. Although Tune-A-Video also targets few-shot scenarios, its generation results are less satisfactory. Based on our experimental results, our method outperforms Tune-A-Video in both objective and subjective metrics.
> 4. The ablation experiments include comparisons with and without the mask-guided motion sequences.
> 5. Furthermore, comparative results with similar controllable video generation methods (e.g., Tune-A-Video and Text-to-Video Zero) are also presented.

---

### Official Review · Reviewer_fSg8 · 2024-11-03

**Soundness:** 2
**Presentation:** 2
**Contribution:** 2
**Rating:** 3
**Confidence:** 4

**Summary:**

This paper proposes the mask-guided video generation to introduce foreground masks for learning region-specific attention. This method first generates the first frame using ControlNet, and allows for incrementally generation of longer video sequences with motion masks conditions.

**Strengths:**

++ The integration of mask condition for masked attention mechanism improves the performance of generated videos.

++ The paper is well written.

**Weaknesses:**

-- The method relies on providing motion masks during inference, which limits its practicality for real-world applications. How to get the motion masks for arbitrary videos? And how robust is the proposed method towards inaccurate masks?

-- The method requires the first frame to be generated first using ControlNet, and then "animate" the first frame with motion mask sequence. However, such a pipeline faces significant challenges to generate videos with complex effects such as changing illuminations, generation with a variable number of subjects, etc. The experiments in this paper also mainly show single subject videos and only one video with multiple birds, especially the single horse running prompt. How could the method be extended to more complex scenarios such as varying illuminations?

-- There is no video results to directly compare the temporal consistency of generated videos. It would be better to provide more video comparisons in supplementary materials.

-- There is no quantitative results in the ablation study, and it remains unclear how many text prompts are used for the ablation study. It is difficult to analyse the effectiveness of the proposed design. Therefore, it is suggested to report quantitative metrics for ablation study such as FID and clearly clarify the number and complexity of prompts used for ablation study.

**Questions:**

Question: Does the model need to be trained separately for each motion pattern?

---

> ### Author Response · Authors · 2024-11-18
>
> 1. In practical applications, some video generation tasks require a control signal (mask) to generate corresponding actions as desired. For instance, an animation director could design specific motion trajectories (e.g., walking, running, or jumping paths) for characters by drawing motion masks, enabling the exploration of different cinematic languages to align with creative visions.
> 2. Our method relies on high-quality motion masks, typically generated through manual annotation or automated tools (such as models like Segment Anything). These methods effectively extract masks for foreground objects.
> 3. The current experiments focus on relatively simple scenarios involving single or multiple objects. Generating complex scenes remains a significant challenge for existing models such as Tune-A-Video, Text2Video-Zero, and LAMP [CVPR 2024]. In future work, we plan to explore using variable lighting conditions as a parameter to replace the mask as the control signal.
> 4. The results of the ablation experiments will be uploaded as supplementary materials.

---

### Official Review · Reviewer_rRB5 · 2024-11-04

**Soundness:** 2
**Presentation:** 3
**Contribution:** 2
**Rating:** 3
**Confidence:** 5

**Summary:**

This paper presented a mask-guided video generation method, which can be trained efficiently on a single GPU. First-frame sharing is adopted to enhance the temporal consistency, while incremental generation is leveraged for generating long videos. Experiments are carried out to evaluate the proposed method.

**Strengths:**

+ The introduction of mask guidance in video generation.
+ First-frame sharing is adopted to enhance the temporal consistency.
+ Incremental generation is leveraged for generating long videos.

**Weaknesses:**

- The technical contribution is not insufficient. Maybe first-frame sharing is new to video generation, but it cannot achieve competing performance in comparison with existing open-sourced video generation models such as CogVideoX, Open Sora, etc. As for mask guidance, similar idea has been suggested in ControlNet for conditional image generation. Incremental generation is also suggested in StreamingT2V [1] StreamingT2V: Consistent, Dynamic, and Extendable Long Video Generation from Text, https://arxiv.org/abs/2403.14773.
- The performance may be inferior to the state-of-the-art video generation methods.

**Questions:**

1. Please compare the proposed method with the state-of-the-art methods such as CogVideoX, Open Sora, etc.
2. Discussion with the related methods.
3. Discussion on the practical value of this work. It seems that better results can be attained by the existing methods.

---

> ### Author Response · Authors · 2024-11-18
>
> 1. Despite the higher performance potential of methods like CogVideoX and Open Sora, these models rely heavily on extensive data and computational resources, making them less suitable for resource-constrained research or application scenarios. In contrast, the method proposed in this study demonstrates the ability to generate high-quality videos even with limited data and a single GPU, offering a lightweight and resource-friendly solution.
> 2. In the experimental section, we also present comparative studies with methods such as Tune-A-Video [ICCV 2023], Text2Video-Zero, and LAMP [CVPR 2024]. The results indicate that our method has certain advantages in terms of consistency and text alignment.
> 3. From a practical perspective, the lightweight nature of our approach makes it well-suited for resource-constrained scenarios. In real-world applications, it is not always feasible to access large datasets or high-performance computing resources. Therefore, the mask-guided video generation method proposed in this study enables training and inference on limited video data and a single GPU, providing an efficient solution for video generation and editing tasks on small-scale devices.

---

### Note · Authors · 2024-11-26

I have read and agree with the venue's withdrawal policy on behalf of myself and my co-authors.